# Regulation of the Gene for Alanine Racemase Modulates Amino Acid Metabolism with Consequent Alterations in Cell Wall Properties and Adhesive Capability in *Brucella* spp.

**DOI:** 10.3390/ijms242216145

**Published:** 2023-11-09

**Authors:** Mingyue Hao, Minghui Wang, Ting Tang, Danyu Zhao, Shurong Yin, Yong Shi, Xiaofang Liu, Gaowa Wudong, Yuanhao Yang, Mengyu Zhang, Lin Qi, Dong Zhou, Wei Liu, Yaping Jin, Aihua Wang

**Affiliations:** 1College of Veterinary Medicine, Northwest A&F University, Yangling District, Xianyang 712100, China; haoshuaifirst@nwafu.edu.cn (M.H.); wangmh@nwafu.edu.cn (M.W.); tangting1829@163.com (T.T.); zhaodanyu@nwafu.edu.cn (D.Z.); yinshurong2442@163.com (S.Y.); hongxuelange@126.com (Y.S.); liuxf@nwafu.edu.cn (X.L.); wudonggaowaa@163.com (G.W.); yangyuanhao3735@163.com (Y.Y.); mengyuzhang@nwafu.edu.cn (M.Z.); qilin2021@nwafu.edu.cn (L.Q.); zhoudong1949@163.com (D.Z.); wliu20cn@yahoo.com (W.L.); yapingjin@nwafu.edu.cn (Y.J.); 2Key Laboratory of Animal Biotechnology of the Ministry of Agriculture, Northwest A&F University, Yangling District, Xianyang 712100, China

**Keywords:** *Brucella suis* S2, *alr*, cell wall, transmission electron microscopy, metabolomics

## Abstract

*Brucella*, a zoonotic facultative intracellular pathogenic bacterium, poses a significant threat both to human health and to the development of the livestock industry. Alanine racemase (Alr), the enzyme responsible for alanine racemization, plays a pivotal role in regulating virulence in this bacterium. Moreover, *Brucella* mutants with *alr* gene deletions (Δ*alr*) exhibit potential as vaccine candidates. However, the mechanisms that underlie the detrimental effects of *alr* knockouts on *Brucella* pathogenicity remain elusive. Here, initially, we conducted a bioinformatics analysis of Alr, which demonstrated a high degree of conservation of the protein within *Brucella* spp. Subsequent metabolomics studies unveiled alterations in amino acid pathways following deletion of the *alr* gene. Furthermore, *alr* deletion in *Brucella suis* S2 induced decreased resistance to stress, antibiotics, and other factors. Transmission electron microscopy of simulated macrophage intracellular infection revealed damage to the cell wall in the Δ*alr* strain, whereas propidium iodide staining and alkaline phosphatase and lactate dehydrogenase assays demonstrated alterations in cell membrane permeability. Changes in cell wall properties were revealed by measurements of cell surface hydrophobicity and zeta potential. Finally, the diminished adhesion capacity of the Δ*alr* strain was shown by immunofluorescence and bacterial enumeration assays. In summary, our findings indicate that the *alr* gene that regulates amino acid metabolism in *Brucella* influences the properties of the cell wall, which modulates bacterial adherence capability. This study is the first demonstration that Alr impacts virulence by modulating bacterial metabolism, thereby providing novel insights into the pathogenic mechanisms of *Brucella* spp.

## 1. Introduction

*Brucella* is a zoonotic facultative intracellular pathogen that is the causative agent of brucellosis infections. The disease elicits chronic debilitation in humans and causes reproductive maladies in animals, with profound repercussions for livestock farming, tourism, and international trade [1,2]. Following cellular entry, *Brucella* induces the formation of *Brucella*-containing vacuoles, thereby sequestering the bacterium from lysosomal engagement and evading the host immune response. The consequent persistent infection is a major challenge in the treatment and prevention of brucellosis [3]. The initial step in *Brucella* infection involves adhesion to host cells, which is a pivotal point in the pathogenic cascade, as adhesion is the primary phase in the infectious cycle of most pathogenic bacteria [4]. Most bacterial pathogens interact with the host by means of adhesive molecules (adhesins) that are displayed prominently on bacterial cell surfaces. In the case of *Brucella*, adhesins are categorized into filamentous (pili) adhesins that comprise complex structures composed of multiple subunits, and non-pili adhesins, which exist as monomeric or trimeric proteins. The majority of bacterial adhesins cluster within the cell wall. Thus, alterations in cell wall properties may trigger corresponding changes in bacterial adhesion capabilities [5].

Alanine racemase (Alr) belongs to the family of isomerases that act upon amino acids and their derivatives and is localized in the cytoplasm. Bacterial Alr catalyzes the conversion of L-alanine into D-alanine, with both isomers serving as crucial constituents of bacterial cell wall peptidoglycans [6]. D-alanine is the most abundant D-type amino acid in bacterial cell wall peptidoglycan chains. Apart from glycine, natural amino acids are categorized into D- and L-type amino acids, with D-type amino acids prevalent across numerous organisms, particularly in bacteria [7]. D-type amino acids participate in the biosynthesis of certain peptides generated through non-ribosomal pathways. Additionally, these amino acids serve as crucial regulators for various bacterial bioprocesses, including the modulation of bacterial adhesion, adjustment of bacterial surface charge, facilitation of cell wall remodeling during the stationary phase, and modulation of virulence in pathogens [8].

Despite extensive research, understanding of the functional repertoire of bacterial Alr remains limited. The current investigation of Alr primarily revolves around validation of the impact of *alr* deletion on cell wall phenotypes and virulence [9]. However, the broader influence of Alr on bacterial metabolism has been overlooked. The correlation between bacterial metabolism and virulence has garnered increased attention recently, which highlights the close relationship between alterations in bacterial metabolic pathways and virulence [10,11].

This study probed the metabolic changes in *Brucella* following *alr* gene deletion, assessed the resistance of the mutant strain to stressors, and examined alterations in cell wall properties due to the absence of *alr*. The adhesive capabilities of the mutant strain were also examined. The results revealed that Alr, through its role in modulating *Brucella* amino acid metabolism, influences surface hydrophobicity, the Zeta potential, and cell wall permeability, thereby impacting the adhesive capacity. This study not only expands our understanding of the multifaceted role of Alr in bacteria but also lays a foundation for further exploration of the pathogenic mechanisms of *Brucella*.

## 2. Results

### 2.1. The Alr Protein Is Highly Conserved in Brucella

All available *Brucella* genome sequences were examined for the *alr* gene, which revealed the universal presence of the locus, including in *Brucella canis* GB1 (C6Y57_05320), *B. suis* 1330 (BS1330_RS14350), *B. suis* S2 (BSS2_RS14335), *B. abortus* 2308 (BAB_RS27865), and *B. melitensis* 16M (DK63_RS03915) (Figure 1A). Comparative analysis of Alr amino acid sequences in *Brucella* S2 showed that the homologs uniformly contained 399 amino acids, with the only variation detected at position 188. Specifically, alanine was present at this position in proteins from *B. canis* GB1, *B. suis* 1330, and *B. suis* S2, whereas valine was encoded at the same position in *B. abortus* 2308 and *B. melitensis* 16M. Nevertheless, the sequence identity in Alr proteins was 99.74%, which underscores the exceptional conservation of the enzyme within the *Brucella* genus.

### 2.2. Physical and Biochemical Properties of the Alr Protein

A comprehensive analysis of the physicochemical properties of the *Brucella* Alr protein was conducted utilizing ProtParam software (https://web.expasy.org/protparam/ accessed on 1 July 2022). The theoretical molecular weight of the protein was determined to be 42,699.07 Da, with a total of 6071 atoms and a molecular formula of C_1911_H_3062_N_528_O_561_S_9_. The isoelectric point (pI) was calculated to be 8.29, which classified Alr as an alkaline protein. The protein consists of 399 amino acids, with alanine and tryptophan comprising the highest (13.3%) and lowest (0.3%) proportions, respectively (Appendix A). The predicted extinction coefficient at 280 nm in an aqueous environment was 26,485 M^−1^ cm^−1^. The half-life of the protein was estimated to be 30 h, with an instability index of 27.2. Additionally, the aliphatic index was determined to be 100.85.

### 2.3. Structure Predictions of Alr

The secondary structure prediction of the Alr protein was conducted using SOMPA web-based software (http://bioinf.cs.ucl.ac.uk/psipred/ accessed on 1 July 2022) (Figure 1B). The predominant secondary structure element was α-helices, which constituted 37.34% of the total amino acids. Extensive β-strands accounted for 19.3%, β-turns for 9.02%, and random coils for 34.34% of the structure of the protein.

The prediction of protein tertiary structures was performed using SWISS-MODEL software (https://swissmodel.expasy.org/interactive accessed on 1 July 2022), which demonstrated a striking degree of structural similarity between the proteins from *B. suis* S2, *B. melitensis* 16M, and *B. abortus* 2308 (Figure 1C), which correlates with the high level of sequence identity.

### 2.4. Numerous Metabolites Are Upregulated and Downregulated in B. suis Δalr through Liquid Chromatography–Mass Spectrometry (LC-MS)

Comparative analysis of the metabolomes between wild-type *B. suis* S2 and Δ*alr* strains lead to the identification of 5841 distinct metabolites, which were categorized based on their proportional quantities as organonitrogen compounds (1.87%), steroids and steroid derivatives (2.52%), prenol lipids (4.38%), glycerophospholipids (4.9%), benzene and substituted derivatives (5.96%), unclassified (7.91%), organooxygen compounds (7.96%), fatty acyls (13.27%), carboxylic acids and derivatives (16.3%), and others (34.94%) (Figure 2A).

Multivariate statistical analyses were conducted, commencing with an unsupervised Principal Component Analysis (PCA) (Figure 2B), to explore the overall distribution of samples and to assess the stability of the entire analytical process. Subsequently, supervised Partial Least Squares Discriminant Analysis (PLS-DA) (Figure 2C) and Orthogonal Partial Least Squares Discriminant Analysis (OPLS-DA) (Figure 2D) were employed to discriminate between the global differences in metabolic profiles of various groups, thereby identifying differentially abundant metabolites. Following univariate analysis, a total of 1545 upregulated metabolites and 1491 downregulated metabolites were identified in *B. suis* Δ*alr* compared to the wild-type strain (Figure 2E).

### 2.5. Differential Metabolite Screening Refines Upregulated and Downregulated Metabolites in B. suis Deleted from the alr Gene

A combined multivariate and univariate analysis approach was employed to screen for differentially abundant metabolites in *B. suis* Δ*alr*. Within the OPLS-DA analysis, the Variable Important in Projection (VIP) score was utilized to gauge the impact strength and discriminatory power of the expression pattern of each metabolite on the classification of samples among metabolite groups. This approach facilitated the identification of biologically significant differential metabolites. Subsequently, a *t*-test was employed to determine the statistical significance of these inter-group differences. The selection criteria for differential metabolites were set as VIP values > 1 for the first principal component of the OPLS-DA model and a *p*-value < 0.05 in the *t*-test. Applying these criteria, we identified a total of 184 upregulated metabolites and 211 downregulated metabolites in *B. suis* Δ*alr* compared to the wild-type strain (Figure 2F).

### 2.6. Differentially Abundant Metabolite Analysis of Diverse Metabolite Comparison Groups

Hierarchical clustering was conducted on the entire set of Differentially Abundant Metabolites (DAMs) in order to provide a more visual representation of the interrelation between samples and the variations in metabolite abundance across distinct samples. The color gradient from blue to red signifies the level of metabolite abundance, with blue indicating lower abundance and red indicating higher abundance of DAMs (Figure 3A). Additionally, we derived the top 50 differential metabolites based on their significance (Figure 3A).

### 2.7. Correlation Analysis of DAMs Reveals Metabolic Changes in B. suis Δalr to Maintain Survival

Correlation analysis serves as a valuable tool for quantifying the associations between significant DAMs and for gaining deeper insights into the interplay among metabolites during biological alterations. Here, Pearson correlation n coefficients were employed to assess the strength of linear relationships between pairs of metabolites. We presented 20 significantly different metabolites, where red signifies positive correlations and blue denotes negative correlations. A deeper color and larger circular area indicate more pronounced differences among the metabolites. Phosphatidylethanolamine (PE) and phosphatidylglycerol (PG) belong to the class of phospholipids, while diglyceride (DG) is classified under neutral lipids. All three are among the primary constituents of bacterial cell membranes, together forming the fundamental structure of these membranes. The primary role of the cell membrane is to maintain the stability of the internal and external cellular environments, regulate the passage of substances, and participate in intercellular interactions. We observed that metabolites associated with the cell membrane are predominantly positively correlated. Ceramide (Cer) is categorized as a sphingolipid and plays a role in regulating bacterial adhesion and invasion. Punicic acid and oxidized glutathione contribute to the regulation of bacterial resistance against oxidative stress. The results indicate that the number of positively correlated DAM pairs and negatively correlated DAM pairs is roughly equal in both *Brucella suis* S2 and Δ*alr* strains. This observation suggests that the absence of the *alr* gene induces a series of metabolic changes in *Brucella*, which contribute to the bacterium’s survival. Moreover, it signifies that the lack of the *alr* gene leads to alterations in the cell membrane, closely associated with bacterial adhesion and antioxidant properties. However, due to the intricate internal mechanisms of bacteria, some metabolites show positive correlations, while others exhibit negative correlations. The specific functions of these correlations will be further validated in subsequent experiments (Figure 3B). 

### 2.8. KEGG Enrichment Analysis Demonstrates Altered Amino Acid Pathways in B. suis Δalr 

Pathway enrichment analysis of DAMs was deployed to assess further the alterations in metabolic pathways that occur following deletion of the *alr* gene in *Brucella*. The most significantly correlated metabolic pathways between the *B. suis* S2 and *Δalr* strains were identified by enrichment analysis of DAMs. The most enriched pathways were cysteine and methionine metabolism, followed by arginine biosynthesis, and then by pantothenate and CoA biosynthesis, glutathione metabolism, and bacterial chemotaxis (Figure 4). Pathways such as alanine, aspartate, glutamate, and biotin metabolism also exhibited substantial differences between wild-type and mutant strains.

### 2.9. Deletion of Alr Reduces Stress Resistance of B. suis S2

The effect of deletion of the *alr* gene on the growth properties of *B. suis* S2 was assessed initially in a chemically defined medium (CDM). Growth of the Δ*alr* mutant was not significantly different compared either to the wild-type strain or a mutant strain complemented with the *alr* gene *in trans* (CΔ*alr*) (Figure 5A). The cell membrane is the primary sensory interface for environmental stressors in bacteria. In view of the altered membrane characteristics of the Δ*alr* strain (see the following sections), we assessed whether deletion of *alr* impacted the stress sensitivity of *B. suis* S2. Survival of the Δ*alr* strain was significantly lower than that of *B. suis* S2 and CΔ*alr* strains under conditions of high osmotic pressure (2% NaCl), which suggests a critical role for *alr* in regulating stress resistance in *B. suis* S2 (Figure 5B). Furthermore, survival of the Δ*alr* strain was notably lower than that of wild-type and CΔ*alr* strains under acidic conditions, which indicates an important function for *alr* in acid stress tolerance (Figure 5C). The viability of the Δ*alr* strain was also significantly reduced upon exposure to H_2_O_2_, which suggests that *alr* also promotes oxidative stress tolerance in *B. suis* S2 (Figure 5E). However, no significant differences in survival rates of the three strains were observed under high-temperature conditions (Figure 5D).

EDTA is a surfactant that damages bacterial cell membranes, leading to death. As the *alr* deletion influenced the membrane characteristics of *B. suis* S2 (see the following sections), we tested the sensitivity of the deletion strain to this compound. The survival of the Δ*alr* strain was significantly lower than that of *B. suis* S2 and CΔ*alr* when exposed to 1 mM EDTA. This reduction persisted from 0.5 to 2 h, and survival of the deletion strain was close to zero at 2.5 and 3 h (Figure 5F). SDS sensitivity testing was conducted as a second approach to assess the membrane characteristics of the Δ*alr* strain. Deletion of *alr* significantly increased the sensitivity of *B. suis* S2 to SDS (Figure 5G), which emphasizes the role of the Alr protein in the maintenance of cell membrane integrity.

The bacterial cell envelope serves both as a barrier to antibiotic penetration and as a target for several antibiotics. Polymyxin B is a peptide antibiotic that interacts strongly with bacterial membranes and is frequently used to assess the membrane characteristics of *Brucella*. Lincomycin is a lincosamide antibiotic that inhibits bacterial protein synthesis by interacting with the ribosomal 50S subunit. The impact of *alr* deletion on the resistance of *B. suis* S2 to polymyxin B and lincomycin was evaluated. Exposure to polymyxin B at concentrations of 600 and 1200 μg/mL produced significantly lower survival rates for the Δ*alr* strain compared to the wild-type and CΔ*alr* strains, which indicated increased sensitivity of the deletion strain to this antibiotic (Figure 5H). *B. suis* S2 and CΔ*alr* exhibited similar resistance levels to lincomycin after pre-treatment with different concentrations of the antibiotic for 30 min. However, the Δ*alr* derivative displayed significantly reduced resistance to lincomycin (Figure 5I). Based on the results of stress assays here, we selected pH 4.5, 1.5 mM H_2_O_2_, and 2% NaCl as conditions that simulate the host intracellular environment for further experiments [12].

### 2.10. Transmission Electron Microscopy Analysis Reveals Cell Wall Defects in the Alr Mutant Strain

The impact of *alr* gene deletion on the cell wall morphology of *B. suis* S2 in the simulated host intracellular environment (pH 4.5, 1.5 mM H_2_O_2_, and 2% NaCl) was assessed via transmission electron microscopy (Figure 6). *B. suis* S2 and the CΔ*alr* strain displayed smooth and tightly connected cell walls. In contrast, the Δ*alr* derivative exhibited irregular cell wall structures and abnormal cell morphology. The Δ*alr* strain also showed signs of cell contraction, uneven cell walls, regions with partial breakage, discontinuities in the cell membrane, and loss of cytoplasmic contents, which indicate significant damage to the integrity of the cell wall. Conversely, *B. suis* S2 and CΔ*alr* demonstrated better tolerance to the intracellular environment, with no apparent damage to the cells (Figure 6).

### 2.11. Deletion of Alr Alters Cell Surface Hydrophobicity

Changes in cell surface properties are an important indicator of bacterial physiological characteristics. Cell surface features influence nutrient absorption, waste metabolism, cell division, and other physiological states and ensure the normal metabolic activities of the cell [13]. The impact of *alr* deletion on the surface properties of *B. suis* S2 was examined by measuring cell surface hydrophobicity (CSH) at different times in the simulated host intracellular environment. There was no significant difference in CSH between Δ*alr* and *B. suis* S2 or CΔ*alr* at time zero (Figure 7A). However, the Δ*alr* mutant exhibited a significant increase in CSH at 12, 24, 36, and 48 h compared to the control strains. These data suggest that Δ*alr* has altered cell adhesion, which leads to a time-dependent increase in CSH.

### 2.12. The Zeta Potential of Wild-Type and Δalr Strains Differs in the Simulated Intracellular Environment

In addition to CSH, the surface charge of cells is an indicator of bacterial growth and replication. The absolute Zeta potential value is directly proportional to the intermolecular forces between bacterial cells, with smaller intermolecular forces making bacterial cells more prone to aggregation [13]. The Zeta potential of wild-type, Δ*alr*, and CΔ*alr* strains decreased in the simulated intracellular environment (Figure 7B). However, the Δ*alr* mutant exhibited a larger change in surface charge compared to wild-type and CΔ*alr* strains both at time zero and at 12, 24, and 36 h. No significant differences between the strains were observed at 48 h. These results suggest that the deletion of *alr* reduces the intermolecular forces between cells, making them more likely to aggregate in the intracellular environment, which ultimately may lead to hindered bacterial growth and replication.

### 2.13. Effects of Intracellular Environment on Membrane Integrity in B. suis Δalr

The preceding experiments demonstrate that deletion of *alr* alters the membrane properties of *Brucella* in simulated intracellular conditions. Therefore, further investigation of the effect of the deletion on membrane permeability was conducted using propidium iodide (PI) staining. PI does not usually stain the *Brucella* chromosome. However, PI may penetrate and stain DNA when the cell membrane is damaged. The resulting fluorescence intensity may be detected using a microplate reader with excitation and emission wavelengths of 535 and 615 nm, respectively. Thus, changes in the intracellular fluorescence intensity of cells in simulated intracellular conditions were monitored to assess the membrane integrity of wild-type, Δ*alr*, and CΔ*alr* strains. The Δ*alr* mutant exhibited a significant increase in intracellular fluorescence compared to *B. suis* S2 and CΔ*alr* at time zero (Figure 7C). This difference became more pronounced with longer incubation times. These results verify that deletion of the *alr* gene causes a loss of cell membrane integrity and that this alteration is more apparent with increased external pressure during prolonged treatment in simulated intracellular conditions.

### 2.14. Changes in Membrane Integrity Observed by Fluorescence Microscopy

PI freely penetrates damaged cell membranes and binds to intracellular DNA, which leads to the emission of red fluorescence. Therefore, the fluorescence microscopy detection of PI staining within cells indirectly reflects changes in the *Brucella* cell membrane [14]. No red fluorescence was observed in wild-type, Δ*alr*, and CΔ*alr* strains without PI treatment. Similarly, very little fluorescence was observed in *B. suis* S2 and CΔ*alr* after 24 h in the simulated intracellular environment, which indicated the absence of significantly damaged cells (Figure 7D). A small amount of fluorescence was observed in the Δ*alr* strain at this timepoint. However, Δ*alr* exhibited a large amount of red fluorescence after 48 h in the simulated intracellular environment, whereas *B. suis* S2 and CΔ*alr* showed only modest fluorescence (Figure 7D). These observations support the suggestion that deletion of *alr* causes the disruption of cell membrane integrity and that longer exposure to the simulated intracellular environment induces more pronounced damage.

### 2.15. The Alr Deletion Induces Release of Membrane-Bound Alkaline Phosphatase and Intracellular Lactate Dehydrogenase

The bacterial cell wall is a distinctive structure that is absent from mammalian cells, which makes the cell wall a common target for clinically relevant antibiotics. Disruption of cell wall synthesis reduces bacterial resistance to stress and may lead to cell rupture and deformation. Alkaline phosphatase (ALP) is a membrane-bound protein in bacteria. ALP activity is undetectable in intact cells but may be evident when the cell is damaged [15]. ALP activity was used here to assess further the effect of the *alr* deletion on membrane integrity (Figure 7E). *B. suis* S2, Δ*alr*, and CΔ*alr* showed no significant differences in ALP activity after 12 h in the simulated intracellular environment. However, the deletion strain exhibited a significant increase in comparison to *B. suis* S2 and CΔ*alr* at 24, 36, and 48 h. These data suggest that the deletion of *alr* increases the permeability of the *Brucella* cell wall and leads to ALP leakage under conditions that simulate the intracellular environment.

Lactate dehydrogenase (LDH) is involved in the final step of glycolysis and thus plays a crucial role in energy metabolism. The enzyme is difficult to detect in the culture medium of intact bacteria. However, LDH may leak when the cell membrane is damaged, which leads to the detection of the enzyme in the extracellular environment [16]. *B. suis* S2, CΔ*alr*, and Δ*alr* showed no differences in extracellular LDH levels at 12 h and 24 h in the simulated intracellular environment (Figure 7F). However, Δ*alr* exhibited a significant increase compared to *B. suis* S2 and CΔ*alr* at 36 and 48 h. Thus, a time-dependent increase in extracellular LDH activity was evident with *B. suis* Δ*alr*. These data further indicate that deletion of the *alr* gene causes damage to the *Brucella* cell membrane, which leads to the leakage of intracellular components.

### 2.16. Deletion of the alr Gene Reduces Host Cell Adhesion

The adhesion of bacteria to host cells is influenced by the properties of the bacterial cell wall, which is a primary factor for intracellular parasitic bacterial infections. Adhesion is generally regarded as an indicator of bacterial virulence. We conducted a combined study using immunofluorescence and bacterial counting methods to investigate the adhesion ability of *Brucella* strains with and without the *alr* gene. RAW264.7 cells are the main target for *Brucella* infection. We first performed immunofluorescence by staining RAW264.7 nuclei in blue with DAPI and *Brucella* in red with an Alexa-Fluor 555-conjugated antibody. It was observed by confocal microscopy that the red fluorescence intensity decreased with the Δ*alr* strain compared to *B. suis* S2 and CΔ*alr* after 1 h of infection, which indicated a significant reduction in the number of bacteria adhering to the host cells (Figure 8A). Additionally, we used bacterial counting methods, which verified that the deletion mutant exhibited a significant decrease in adhesion rates at 5, 10, 15, 30, 45, and 60 min compared to *B. suis* S2 and CΔ*alr* (Figure 8B).

*Brucella* primarily invades the reproductive and immune systems of the host after the initial infection phase. Bovine endometrial epithelial cells (BEEC) are non-phagocytic cells that are one of the main targets for *Brucella* infection. We DAPI-stained BEEC nuclei in blue and *Brucella* cells in red with the Alexa-Fluor 555-conjugated antibody and examined bacterial adhesion via confocal microscopy. The red fluorescence intensity decreased significantly with the Δ*alr* mutant compared to wild-type and CΔ*alr* strains after 1 h of infection, which indicated a significant reduction in the number of bacteria adhering to the BEEC (Figure 9A). Moreover, bacterial counting showed that Δ*alr* exhibited a significant decrease in adhesion rates compared to *B. suis* S2 and CΔ*alr* at 1, 2, 3, and 4 h (Figure 9B). In summary, the data reveal that deletion of the *alr* gene reduced the adhesion of *Brucella* to both RAW264.7 and BEEC cell lines. 

## 3. Discussion

To probe the metabolic changes in *Brucella* following *alr* gene deletion, we first conducted a bioinformatics analysis to ascertain the level of conservation of the Alr protein within *Brucella* species. Additionally, we elucidated fundamental characteristics of the enzyme, including amino acid composition and structure. Deletion of the *alr* gene resulted in the upregulation of 184 metabolites and the downregulation of 211 metabolites in *B. suis* S2. Moreover, KEGG pathway analysis revealed that Alr primarily regulates amino acid metabolism pathways, which are closely associated with *Brucella* cell wall synthesis and virulence. We observed decreased resistance to diverse stress factors in *Brucella* following *alr* gene deletion. Moreover, cells with the deletion exhibited significant cell wall damage and pronounced alterations in cell membrane permeability in simulated intracellular conditions. These changes included reduced CSH and diminished Zeta potential, which lead to the inferior adhesion capability of *Brucella* to host cells.

Sequence homology alignment of the Alr protein revealed that, apart from position 188, the sequences were entirely conserved across five diverse *Brucella* species. Additionally, we found that the tertiary structures of the homologs were largely congruent. These findings underscore the high degree of conservation of the Alr protein within the *Brucella* genus. Similarly, the Alr proteins from *Burkholderia pseudomallei* (BPSL2179 and BPSS0711), *B. mallei* (BMA1575), *B. thailandensis* (BTH_I2007 and BTH_II1715), *Escherichia coli* (EcoliDadX and EcoliAlr), and *Pseudomonas aeruginosa* (PAO1DadX and PAO1Alr) exhibited similarity ranging from 60% to 99% [17]. This suggests minimal compositional differences in the Alr protein across diverse bacterial species, thereby enhancing the prospects of developing broad-spectrum drugs that target Alr [18].

The synthesis of bacterial cell walls requires multiple amino acids, including d-Ala, d-Glu/Gln, d-Asp, d-Met, d-Trp, and d-Phe [19,20]. Our results indicate that the deletion of *alr* in *B. suis* leads to alterations in amino acid metabolism pathways, including alanine, aspartate, glutamate, glycine, serine, and threonine metabolism. Certain metabolites were upregulated whereas others were downregulated in the mutant strain, which suggests that *Brucella* employs specific compensatory mechanisms to maintain normal amino acid biosynthesis when particular genes are compromised. Our findings also revealed changes in pathways related to cysteine, methionine, and pyruvate metabolism. Previous studies, including with *Citrobacter rodentium*, have demonstrated that changes in arginine metabolism may affect intracellular survival and adhesion capability [21]. Additionally, the cyclopropane–fatty acid–acyl-phospholipid synthase (*cfa*) gene regulates cysteine metabolism and impacts phospholipid content, hydrophobicity, and fluidity, which promotes alterations in the membrane lipid composition [22,23]. Furthermore, alanine and glutamate are upstream metabolites in pyruvate metabolism, which in turn regulate the tricarboxylic acid cycle. Thus, the deletion of *alr* in *Brucella* not only affects the amino acid composition but also is accompanied by energy-related changes [24].

The function of the Alr protein in bacteria has been limited to date to the interconversion of L- and D-alanine isomers. However, deletion of the *alr* gene results in cell wall damage and enhanced cell membrane permeability in *Aeromonas hydrophila* [25]. In our study, we observed a reduction in stress resistance in *Brucella* following *alr* deletion. Transmission electron microscopy revealed structural disruptions in the *Brucella* cell wall that were accompanied by the leakage of cytoplasmic contents. Similar results were observed in *E. coli* following *alr* gene deletion [26]. Furthermore, under simulated intracellular conditions, PI was able to penetrate the cell wall of the *alr*-deficient mutant, bind to intracellular DNA, and emit fluorescence. Significant increases in intracellular ALP and LDH levels were detected in the supernatant of the deletion strain. These observations suggest a change in membrane permeability in *Brucella* following *alr* deletion, as ALP and LDH activities are not typically detectable in bacterial cultures [27].

Mutation of the *alr* gene in *Streptococcus mutans* induced alterations in biofilm biomass, biofilm structure, extracellular polysaccharide (EPS) synthesis, glucosyltransferase (*gtf*) gene expression, acid production, and acid tolerance, although bacterial-adhesion capability was not examined [10]. Similarly, changes in cell wall properties were associated with alterations in the adhesion capability [8]. CSH refers to the redistribution of bacteria at the water–oil interface under various forces and is a crucial factor in controlling bacterial growth [28]. In polar aqueous systems, higher CSH makes bacteria more prone to aggregation, thereby impacting normal growth. The Zeta potential refers to the electric potential at the liquid–solid layer on bacterial cell surfaces, and changes in surface charge can be detected by measuring the Zeta potential [29,30]. Changes in either CSH or the Zeta potential may influence bacterial adhesion. We observed a significant increase in cell wall CSH and a marked decrease in the Zeta potential in the *B. suis alr* mutant compared to the wild-type strain. These results collectively indicate that *alr* deletion allows the penetration of external substances to the cell, which promotes increased leakage of intracellular substances.

*Brucella* is an intracellular parasitic bacterium that is difficult to eradicate via the host immune system or using exogenous drugs. The adhesion process is crucial to the pathogenic mechanism of *Brucella*. The bacterium replicates within professional phagocytic cells, including macrophages, dendritic cells, and granulocytes, as well as non-professional phagocytic cells, including epithelial cells, fibroblasts, and nourishing cells [31,32]. In this study, we assessed the adhesion capability of *Brucella* to phagocytic cells (RAW264.7) and non-phagocytic cells (BEEC) using immunofluorescence assays and bacterial-enumeration techniques, which revealed a significant decrease in adhesion capability in *Brucella* after *alr* deletion. As adhesion is a critical step in cell infection and an essential component of virulence for intracellular parasites such as *Brucella*, the reduced adhesion capability induced by *alr* deletion implies a decrease in the virulence of *Brucella*.

In summary, this study demonstrated that the deletion of *alr* in *Brucella* primarily induced alterations in amino acid metabolism, with consequent effects on the properties of the cell wall that lead to reduced host cell adhesion. Our findings indicate for the first time that *alr* modulates bacterial virulence through the regulation of metabolism, thereby offering a novel perspective for elucidating the pathogenic mechanisms of *Brucella*. Furthermore, these insights provide valuable clues for the development of genetic engineering-based vaccines that target Alr and the pathways that this protein modulates.

## 4. Materials and Methods

### 4.1. Bioinformatics Analysis of the Alr Protein

The investigation employed the Alr amino acid sequence from *B. suis* S2 (Gene ID: BSS2_RS14335) as the primary subject of analysis. Sequence comparison was executed through utilization of the BLAST function accessible on the National Center for Biotechnology Information (NCBI) website. Subsequently, the prediction of the amino acid sequence of the Alr protein, as well as the evaluation of its extinction coefficient and the assessment of its physical and chemical stability, were accomplished using the online software ProtParam (https://web.expasy.org/protparam/ accessed on 1 July 2022). Prediction of the Alr protein secondary structure was conducted with SOMPA software [33], accessible via http://bioinf.cs.ucl.ac.uk/psipred/ (accessed on 1 July 2022). Prediction of protein tertiary structure was performed with SWISS-MODEL software https://swissmodel.expasy.org/interactive/ (accessed on 1 July 2022).

### 4.2. Biosafety Statement

Experiments were conducted according to the “Regulations on Biosafety of Pathogenic Microorganism Laboratory” (2004) No. 424 prescribed by the State Council of the People’s Republic of China and were approved by the Biosafety Committee of Northwest A&F University.

### 4.3. Bacterial Strains

*B. suis* S2 wild-type strain with CVCC reference number CVCC70502 was obtained from the Shaanxi Provincial Institute for Veterinary Drug Control, Xi’an, Shaanxi, China. The Δ*alr* and CΔ*alr* strains were constructed previously in our laboratory [34]. All *Brucella* strains were cultivated in CDM. Tryptic Soy Agar (TSA) medium was used for plate-counting purposes. A description of the constituents of CDM is provided in Appendix A.

### 4.4. Sample Preparation for Liquid Chromatography–Mass Spectrometry

Nine cultures each of *B. suis* S2 and Δ*alr* were grown in TSA medium to OD600 of approximately 0.6. Cultures were washed three times with pre-chilled sterile physiological saline solution, followed by centrifugation at 10,000 rpm for 10 min. A total of 30 milligrams of each sample was combined with 20 μL of an internal standard and 1 mL of methanol water following the protocol established by Wu et al. [35]. Subsequently, two small steel balls were introduced into the mixture, which was then pre-cooled at −20 °C for 2 min prior to grinding (60 Hz, 2 min). Ultrasonic extraction was performed in an ice water bath for 30 min, followed by overnight storage at −20 °C. The mixture was centrifuged for 10 min at 13,000 rpm at 4 °C, and 150 μL of supernatant was withdrawn using a syringe. This supernatant was designated as the organic phase, which was filtered through a 0.22 μM pinhole filter before transfer into a liquid chromatography injection vial, which was stored at −80 °C prior to liquid chromatography–mass spectrometry (LC–MS) analysis.

### 4.5. LC–MS and Data Processing

The analytical instrumentation employed comprised an LC–MS system that consisted of a Nexera UPLC ultrahigh-performance liquid chromatograph coupled with a QE high-resolution tandem mass spectrometer (Shimadzu Corporation, Tokyo, Japan). The chromatographic conditions were configured as follows: the column was an ACQUITY UPLC HSS T3 (100 mm × 2.1 mm, 1.8 μm); the mobile phase consisted of A, which was water containing 0.1% formic acid, and B. acetonitrile containing 0.1% formic acid. The flow rate was maintained precisely at 0.35 mL/min. MS analysis was executed utilizing an ESI ion source with sample mass spectrum signal acquisition conducted in both positive and negative ion scanning modes. For qualitative analysis, the PMDB database was consulted, with the analytical procedures following the methodologies established by Li et al. [36]. Quantitative analysis was conducted in accordance with the methodologies outlined by Cao et al. [37].

### 4.6. LC–MS Data Statistical Analysis

Multivariate statistical analysis encompassed a repertoire of techniques, including PCA, PLS-DA, and OPLS-DA. In contrast, univariate analysis primarily centered on univariate description and statistical inference, which encompassed interval estimation and the conduction of statistical hypothesis testing. Comparative assessments of metabolites between two groups were undertaken employing Student’s *t*-tests and fold change analyses. Furthermore, the amalgamation of multidimensional and unidimensional analyses was deployed to identify DAMs across the groups. The selection criteria consisted of a Variable Importance in Projection (VIP) value >1 for the first principal component of the OPLS-DA model and a *p*-value derived from a *t*-test < 0.05. A comprehensive metabolic pathway enrichment analysis of DAMs was executed utilizing the KEGG database in accordance with the approach elucidated by Wu et al. [38]. For investigations pertaining to different shading treatments, one-way analysis of variance was conducted, followed by a post hoc Duncan’s multiple range test. Statistical analysis for this study was conducted using R software (version 3.6.2), encompassing various multivariate analyses like PCA, PLS-DA, and more, all facilitated through the “ropls” package within R. Additionally, we utilized specific packages, such as “pheatmap”, “ggplot2”, and “ggrepel” for data visualization, as well as “corrplot” for correlation analysis. The significance of our findings was determined using our analytical approaches, and results with *p* values < 0.05 were considered statistically significant.

### 4.7. Analysis of the Growth of B. suis Strains

*B. suis* strains were inoculated in 10 mL of CDM at equivalent densities (1 × 10^7^ colony-forming units (CFU)) and were incubated at 37 °C with continuous agitation. The cultures were harvested at 0, 8, 16, 24, 32, 40, 48, 56, 64, 72, and 80 h, and the OD_600_ was determined within a 96-well plate utilizing a microplate reader (Bio-Rad, California, USA). At the same time, we employed the double-dilution method to enumerate CFU at each time point (Appendix A).

### 4.8. Brucella Stress Resistance Tests

*Brucella* strains were inoculated into CDM containing 2% NaCl, and OD_600_ values were measured at eight-hour intervals. The final measurement was taken at 96 h, and the data were used to construct growth curves. Additionally, *Brucella* strains were inoculated into CDM with different pH levels (3.5 and 4.5) and cultured for 0.5 h. Strains were also subjected to heat stress at 42 °C in CDM for 0.5, 1, and 2 h. Furthermore, the strains were grown in CDM containing H_2_O_2_ (1.0 and 2.5 mM) for 1 h and in medium with EDTA (1 mM) for 0.5, 1, 1.5, 2.5, and 3.0 h. Following the various treatments, each sample was serially diluted ten-fold and plated onto TSA. Plates were incubated at 37 °C for three days, and the survival of each strain was calculated by dividing the CFU of the treated bacteria by the CFU of the untreated samples. For the SDS sensitivity assay, *Brucella* strains (10^4^–10^6^ CFUs /mL) were subjected to tenfold dilution, and 10 μL of the diluted samples was inoculated onto TSA plates containing 0.00325% SDS. The plates were incubated at 37 °C for two days. For antibiotic resistance testing, cultures were placed in a 96-well enzyme-linked immunosorbent assay (ELISA) plate (at 2 × 10^4^ CFUs/well) and were exposed to polymyxin B at concentrations of 1200, 600, and 300 μg/mL or lincomycin at concentrations of 500, 375, and 250 μg/mL at 37 °C for 1 h. The cultures were tenfold diluted and plated onto TSA agar, followed by incubation at 37 °C for 3 days. The CFUs after antibiotic treatment were counted and compared to those of untreated cultures to calculate the survival rate for each sample [14].

### 4.9. Transmission Electron Microscopy

Bacterial cultures in logarithmic growth phase were centrifuged at 5000× *g* for 8 min and were then subjected to triple washes with PBS. Subsequently, the suspension was mixed with PBS (pH 4.5) containing 1.5 mM H_2_O_2_ and 2% NaCl and incubated for 12, 24, and 48 h with corresponding control samples. Samples were treated initially with a 3% glutaraldehyde solution, followed by postfixation in 1% osmium tetroxide. Dehydration was performed using a series of acetone solutions, and subsequent infiltration was achieved with Epox 812 resin for an extended period. The specimens were subsequently embedded. Semithin sections were stained with methylene blue, whereas ultrathin sections were produced using a diamond knife and further stained with uranyl acetate and lead citrate. These sections were examined using a JEM-1400-FLASH Transmission Electron Microscope (JEOL, Tokyo, Japan).

### 4.10. Determination of Cell Surface Hydrophobicity

CSH was assessed utilizing the carbon–hydrogen adsorption capacity method [39]. Logarithmic-phase bacterial cultures were centrifuged at 8000 rpm, washed with PBS in triplicate, and adjusted to OD_600_ = 0.5. Subsequently, 4 mL of PBS (pH 4.5) containing 1.5 mM H_2_O_2_ and 2% NaCl was mixed with the suspension and incubated for 12, 24, 36, and 48 h, with corresponding control samples. A total of 1 mL of hexadecane (Tianjin KeMiOu Chemical Reagent Co., Ltd., Tianjin, China) was added to each sample. After vortexing for 1 min, the samples were left undisturbed at room temperature for 15 min to allow for complete separation of the aqueous and organic phases. The aqueous phase was extracted, and the OD_600_ value was recorded. CSH was calculated as CSH (%) = (OD_0_ − OD_1_)/OD_0_ × 100%, where OD_0_ represents the OD value before extraction and OD_1_ corresponds to the OD value after extraction.

### 4.11. Measurement of Zeta Potential

Bacterial cultures were grown to logarithmic phase, and 4 mL of PBS (pH 4.5) containing 1.5 mM H_2_O_2_ and 2% NaCl was added to bacteria after centrifugation. The mixtures were then incubated for 12, 24, 36, and 48 h with corresponding control samples. Samples were centrifuged at 10,000 rpm, the supernatant was discarded, and the pellet was washed three times with PBS and then resuspended in PBS. The surface charge was determined using a ZEN3600 Zeta potential analyzer (Nano Laser Particle Size Analyzer, Malvern Instruments Ltd., Malvern, UK). Each experiment was replicated three times, and the results were averaged.

### 4.12. Alkaline Phosphatase Assay

Assessment of ALP leakage from *Brucella* cells was carried out using the ALP assay kit (Beyotime, Shanghai, China). Bacterial cultures in logarithmic growth phase were centrifuged at 5000 rpm for 8 min. The pellets were washed three times with PBS. Subsequently, the bacterial suspensions were mixed with PBS (pH 4.5) containing 1.5 mM H_2_O_2_ and 2% NaCl and incubated for 12, 24, 36, and 48 h. OD_510_ measurements were made.

### 4.13. Lactate Dehydrogenase Assay

The leakage of LDH from *Brucella* strains was assessed using the LDH assay kit (Beyotime). Cultures in logarithmic growth phase were centrifuged at 8000 rpm for 10 min. The pellets were washed three times with PBS and then adjusted to OD_600_ = 0.5. PBS (pH 4.5) containing 1.5 mM H_2_O_2_ and 2% NaCl was added to the bacterial suspensions, followed by incubation for 12, 24, 36, and 48 h. Absorbance values were measured at 490 nm.

### 4.14. Determination of Membrane Permeability

PBS (pH 4.5) containing 1.5 mM H_2_O_2_ and 2% NaCl was added to bacterial cultures in logarithmic growth phase. The mixtures were incubated for 12, 24, 36, and 48 h, followed by centrifugation at 8000 rpm for 10 min, washing, and resuspension in PBS. Subsequently, 2.9 μmol/L PI was added to the sample, which was incubated in the dark at 37 °C for 15 min. The suspension was centrifuged again for 10 min, followed by three washes with PBS. Fluorescence intensity was measured with excitation set at 495 nm and emission set at 615 nm.

### 4.15. Fluorescence Microscopy

Bacterial cultures in logarithmic growth phase were supplemented with PBS (pH 4.5) containing 1.5 mM H_2_O_2_ and 2% NaCl and incubated for 24 and 48 h. The suspensions were centrifuged at 8000 rpm for 10 min and, after washing and resuspension in PBS, were treated with 2.9 μmol/L PI. The samples were incubated in the dark at 37 °C for 15 min. The samples were centrifuged for 10 min and subjected to three PBS washes. Finally, the samples were observed using an inverted fluorescence microscope.

### 4.16. Immunofluorescence Assay

RAW264.7 and BEEC cell lines obtained from the National Collection of Authenticated Cell Cultures were cultivated in Roswell Park Memorial Institute 1640 medium (Hyclone Laboratories Inc., Logan, UT, USA) supplemented with 10% fetal bovine serum (FBS; Hyclone) and were maintained at 37 °C under 5% CO_2_. RAW264.7 and BEEC cells were seeded into 12-well plates and subsequently subjected to infection with *B. suis* strains at a multiplicity of infection of 200. Following infection, the cells were washed twice with PBS at the 1 h post-infection timepoint. Subsequently, the cells were fixed with 4% paraformaldehyde at room temperature for 30 min. Samples were washed in triplicate with PBS and then incubated in PBS containing 0.25% Triton X-100 at room temperature for 20 min. The primary antibody employed was goat anti-*Brucella* polyclonal antibody (diluted 1:1000; ThermoFisher, Waltham, MA, USA), followed by the application of donkey anti-goat Alexa Fluor 555 (ThermoFisher) as the secondary antibody (diluted 1:1000). Nuclei were counterstained with DAPI for 10 min. The slides were washed with PBS four times after each incubation step. Images were acquired using an A1R confocal microscope (Nikon, Tokyo, Japan).

### 4.17. Enumeration of B. suis S2 in Infected RAW264.7 Cells

RAW264.7 cells that had been subjected to infection were washed with PBS at 5, 10, 15, 30, 45, and 60 min post-infection. BEEC cells that had undergone infection were washed with PBS at 1, 2, 3, and 4 h post-infection. Infected cells were lysed utilizing PBS supplemented with 0.5% Triton X-100 for 12 min. The lysates were diluted in PBS, plated onto TSA, and incubated at 37 °C for 72 h. CFUs were enumerated. The percentage of attached bacteria was calculated as follows: attached bacteria % = (CFU of attached bacteria)/(initial number of infecting bacteria) × 100.

### 4.18. Statistical Analysis

Statistical analysis was performed with GraphPad Prism software version 8 (GraphPad Software Inc., La Jolla, CA, USA). The significance of the results was determined using one-way or two-way ANOVA. *p*-values < 0.05 were considered statistically significant.

## Figures and Tables

**Figure 1 ijms-24-16145-f001:**
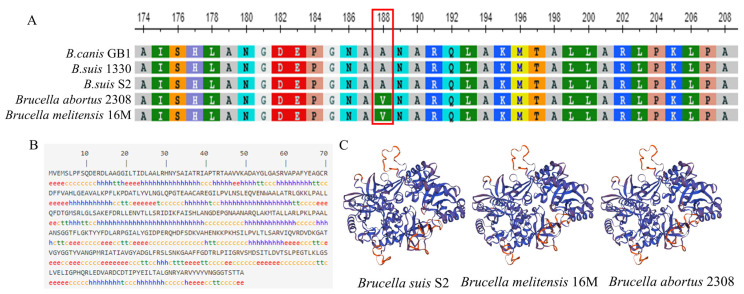
(**A**). Alignment of Alr amino acid sequences from *B. canis* GB1 (AVO71293.1), *B. suis* 1330 (WP_004690358.1), *B. suis* S2 (WP_004690358.1), *B. abortus* 2308 (WP_002967219.1), and *B. melitensis* 16M (WP_002967219.1). Amino acid differences at position 188 are highlighted in red boxes. (**B**). Prediction of the secondary structure of the Alr protein from *B. suis* S2. (**C**). SWISS-MODEL software (https://swissmodel.expasy.org/interactive accessed on 1 July 2022) prediction of the tertiary structures of the Alr proteins from *B. suis* S2, *B. abortus* 2308, and *B. melitensis* 16M.

**Figure 2 ijms-24-16145-f002:**
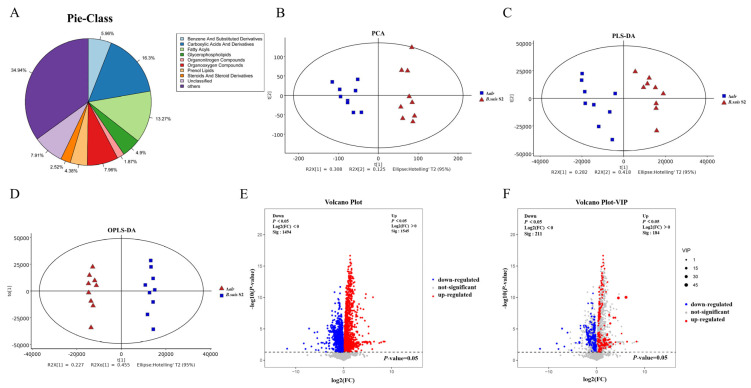
Metabolite classification of *B. suis* S2 and differential metabolite screening using Multivariate Analysis Methods. (**A**). Metabolite classification, with different colors representing different categories of metabolites. (**B**). Unsupervised PCA. (**C**). Supervised PLS-DA. (**D**). OPLS-DA. Blue boxes and red triangles represent *B. suis* S2 and Δ*alr* samples, respectively. (**E**). Univariate statistical analysis. (**F**). Differential metabolite screening. Blue indicates downregulated metabolites, red denotes upregulated metabolites, and gray represents metabolites with no significant differences in the Δ*alr* strain compared to the wild-type strain.

**Figure 3 ijms-24-16145-f003:**
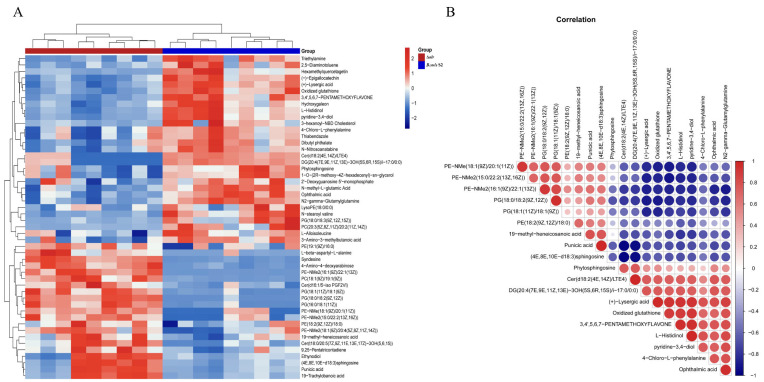
Heatmap of differential metabolites enriched in *B. suis* S2 and Δ*alr* and correlation analysis of differentially enriched metabolites. (**A**). Heatmap of the top 50 differential metabolites. The horizontal axis represents sample names, whereas the vertical axis represents differentially enriched metabolites. (**B**). Correlation analysis of differential metabolites. Red and blue indicate positive and negative correlations, respectively.

**Figure 4 ijms-24-16145-f004:**
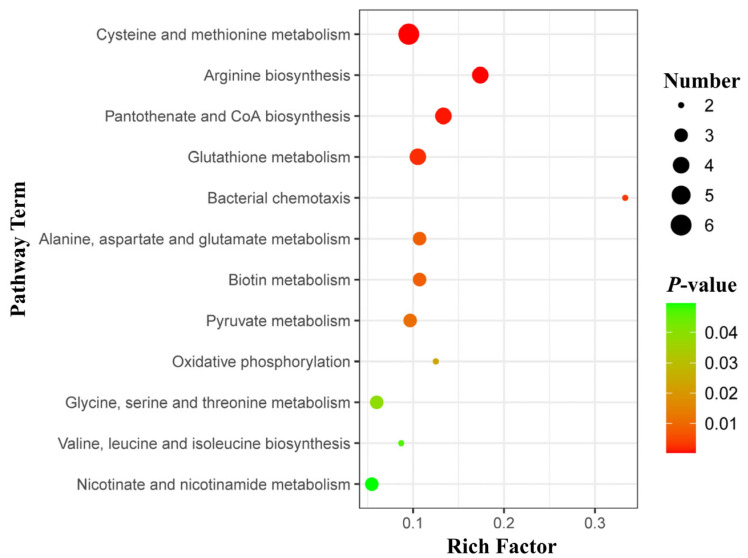
Bubble diagram of DAMs in the top 20 metabolic pathways. The ordinate shows the names of the metabolic pathways, and the abscissa is the enrichment factor (rich factor = number of significant DAMs/number of total metabolites in the pathway).

**Figure 5 ijms-24-16145-f005:**
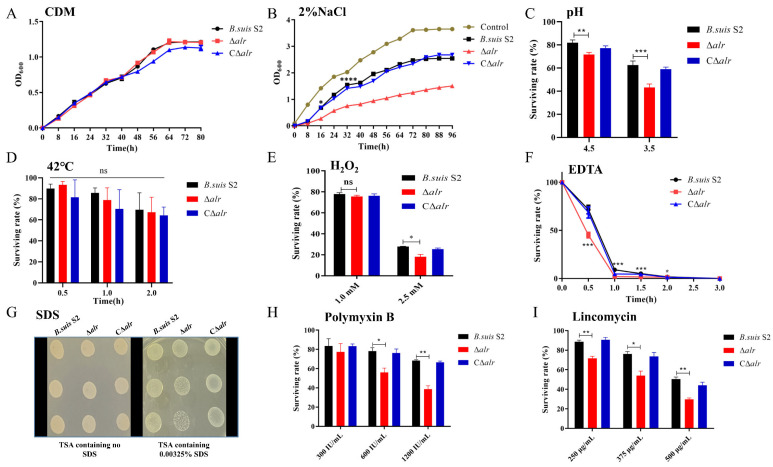
Effects of *alr* deletion on growth characteristics and stress resistance of *B. suis* S2. (**A**). Growth patterns of wild-type, Δ*alr*, and CΔ*alr* strains in CDM. (**B**). Growth curves of wild-type, Δ*alr*, and CΔ*alr* strains in CDM with 2% NaCl. (**C**). Survival of wild-type, Δ*alr*, and CΔ*alr* strains in CDM at pH 4.5 and pH 3.5. (**D**). Survival of wild-type, Δ*alr*, and CΔ*alr* strains in CDM at 42 °C. (**E**). Survival of wild-type, Δ*alr*, and CΔ*alr* strains in CDM with 1.0 and 2.5 mM H_2_O_2_. (**F**). Survival of wild-type, Δ*alr*, and CΔ*alr* strains in CDM with 1 mM EDTA. (**G**). Sensitivity of *B. suis* S2 Δ*alr* to SDS. Ten microliters of *Brucella* samples was added to a TSA plate containing 0.00325% SDS or without SDS and incubated for 48 h at 37 °C. (**H**). Survival of wild-type, Δ*alr*, and CΔ*alr* strains in CDM with 300, 600, and 1200 μg/mL polymyxin B. (**I**). Survival of wild-type, Δ*alr*, and CΔ*alr* strains in CDM with 250, 375, and 500 μg/mL lincomycin. All data are presented as mean ± SEM from three independent experiments. * *p* < 0.05, ** *p* < 0.01, *** *p* < 0.001, **** *p* < 0.0001; ns, not significant.

**Figure 6 ijms-24-16145-f006:**
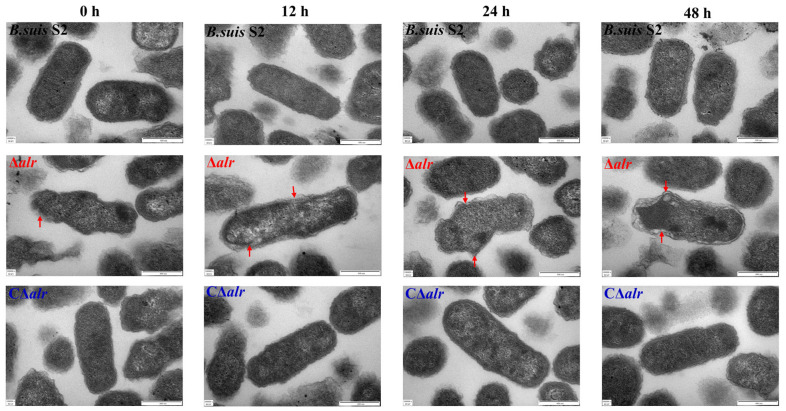
Impact of *alr* gene deletion on cell wall structure of *B. suis* S2 under simulated intracellular conditions. Observation of cell wall structural changes in *Brucella* at 0 h, 12 h, 24 h, and 48 h by transmission electron microscopy. Red arrows indicate areas of cell wall damage.

**Figure 7 ijms-24-16145-f007:**
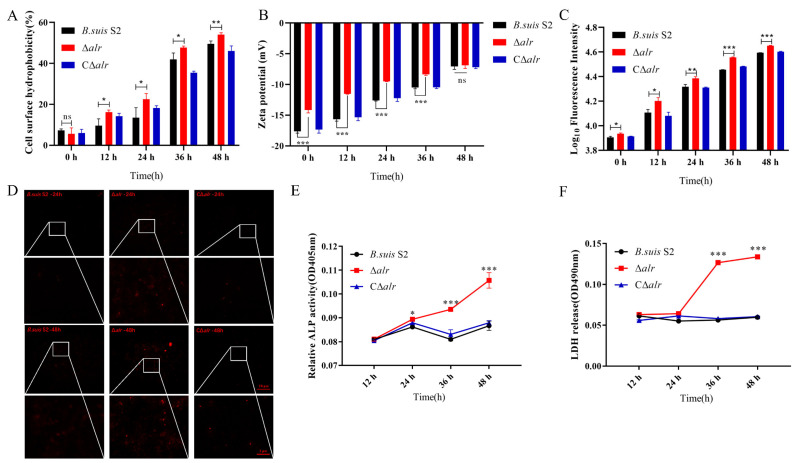
Changes in cell wall properties of *B. suis* S2 deleted from the *alr* gene under simulated intracellular conditions (PBS (pH of 4.5) with 1.5 mM H_2_O_2_ and 2% NaCl). (**A**) Detection of surface hydrophobicity of wild-type, Δ*alr*, and CΔ*alr* strains at 0, 12, 24, 36, and 48. (**B**) Measurement of Zeta potential of wild-type, Δ*alr*, and CΔ*alr* strains at 0, 12, 24, 36, and 48 h. (**C**) Fluorescence intensity of wild-type, Δ*alr*, and CΔ*alr* samples containing the same number of cells. Samples containing 2 × 10^7^ CFUs each were taken and stained with PI, and the fluorescence intensity of each sample was measured at 0, 12, 24, 36, and 48 h. (**D**) Detection of cell wall permeability at 24 and 48 h using inverted fluorescence microscopy. (**E**) Measurement of extracellular alkaline phosphatase activity in wild-type, Δ*alr*, and CΔ*alr* strains at 12, 24, 36, and 48 h. (**F**) Measurement of extracellular LDH activity in wild-type, Δ*alr*, and CΔ*alr* strains at 12, 24, 36, and 48 h. All data are presented as mean ± SEM from three independent experiments. * *p* < 0.05, ** *p* < 0.01, *** *p* < 0.001; ns, not significant.

**Figure 8 ijms-24-16145-f008:**
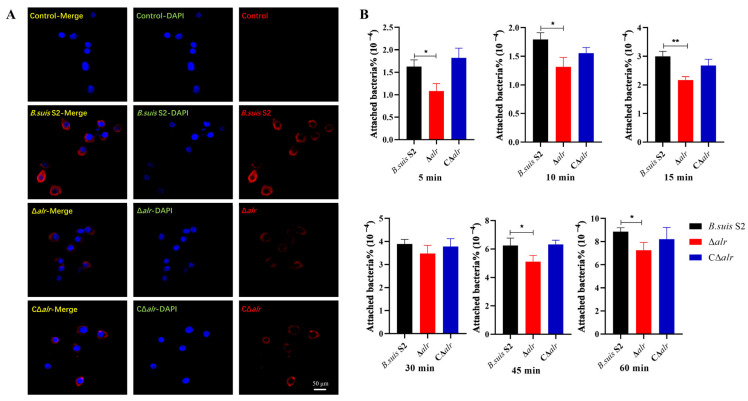
Impact of *alr* gene deletion on the adhesion ability of *B. suis* S2 to RAW264.7 cells. (**A**) Immunofluorescence detection of wild-type, Δ*alr*, and CΔ*alr* strains after 1 h of infection (infection ratio of 200:1). (**B**) Enumeration of adherent wild-type, Δ*alr*, and CΔ*alr* strains at 5, 10, 15, 30, 45, and 60 min using the plate-counting method. All data are presented as mean ± SEM from three independent experiments. * *p* < 0.05, ** *p* < 0.01.

**Figure 9 ijms-24-16145-f009:**
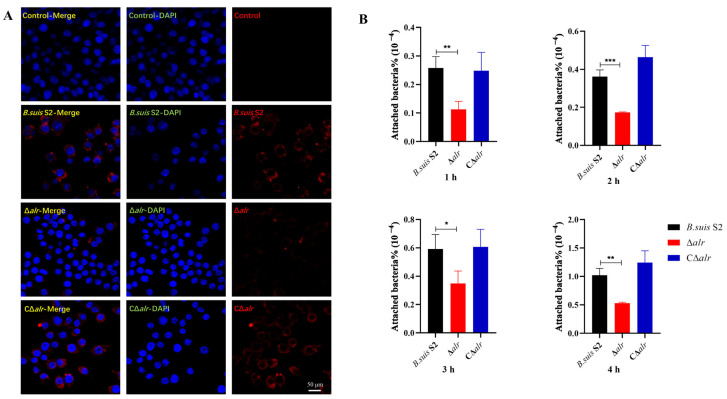
Impact of *alr* gene deletion on the adhesion of *B. suis* S2 to BEEC. (**A**) Immunofluorescence detection of adhesion of wild-type, Δ*alr*, and CΔ*alr* strains after 1 h of infection (infection ratio of 200:1). (**B**) Enumeration of adherent wild-type, Δ*alr*, and CΔ*alr* strains at 1, 2, 3, and 4 h using the plate-counting method. All data are presented as mean ± SEM from three independent experiments. * *p* < 0.05, ** *p* < 0.01, *** *p* < 0.001.

## Data Availability

Data is contained within the article.

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
