# Peer review of "Regulation of the Gene for Alanine Racemase Modulates Amino Acid Metabolism with Consequent Alterations in Cell Wall Properties and Adhesive Capability in Brucella spp."

_ijms, 2023, doi:10.3390/ijms242216145_

Round 1

Reviewer 1 Report

Comments and Suggestions for Authors

In this manuscript the authors, Hao et al. successfully showed the effect of Alr protein in the pathogenesis of Brucella suis S2. At first they demonstrated that Alr is a highly conserved protein in Brucella sp. Then they investigated how Alr protein regulates the pathogenesis of Brucella sp. They made a mutant with alr gene deletion in B. suis S2 and showed that this mutant had changes in cell wall properties, alteration in cell membrane permeability which eventually led defective adhesion of Brucella sp in host cell. The experimental design, presentation of the results and the interpretation of the data are satisfactory. The specific comments for this manuscript are as below:

Major Comments:

1.  In figure 5 where authors presented the bacterial growth curve with different treatment, they used OD600 in Y-axis and time in X-axis. Did the authors do the actual bacterial CFU count at each time point? If yes they should show the same graph but with actual number of CFU in Y-axis and time in X-axis in supplement figure. If they did not count the CFU in each time point, then the authors should do one representative assay where they can corelate the OD with the actual CFU and put that as a supplement figure in revised manuscript.

2. The authors should give a brief introduction about how defect in the cell wall and and cell membrane of Brucella led to the altered pathogenesis in host.

Minor comments:

1. Figure 2: The markings (text) in the figure should be improved for better visualization.

2. Figure 5: In the graph A and B the treatment should be mentioned on the top of the graph. In the graph E, H and I the time of incubation should be mentioned on the top of the graph. In graph B 'control' should be defined in figure legend

3. Line 262: Briefly define 'the stimulated host intracellular environment'.

4. Figure 7: Graph C should be expressed in bar diagram as Graph A and B are in bar diagram.

5. Line 562: If the CFU count is added in the figure, the method for CFU count should be described in method section.

6. Line 566: Please provide a reference of 'stress resistance test' protocol.

Author Response

Dear reviewer,

We are delighted to submit our revised manuscript titled " Regulation of the Gene for Alanine Racemase Modulates Amino Acid Metabolism with Consequent Alterations in Cell Wall Properties and Adhesive Capability in Brucella spp." for your consideration. We appreciate the insightful feedback provided by the reviewers and the opportunity to enhance our work based on your valuable comments. In this manuscript, we have systematically explored the impact of the Alr protein on the pathogenesis of Brucella suis S2. Our study initially established the high conservation of Alr within the Brucella genus, followed by a comprehensive investigation into the regulatory role of Alr in Brucella pathogenesis. Notably, we created a mutant strain with the alr gene deleted in B. suis S2, revealing profound changes in cell wall properties, alterations in cell membrane permeability, and consequent defects in Brucella adhesion to host cells. The study's experimental design, presentation of results, and data interpretation have all been thoroughly scrutinized and refined to meet the highest standards. Please find below our responses to the specific comments from the reviewers, detailing the improvements we have made to address their suggestions. We are confident that the revised manuscript now meets the rigorous standards of International Journal of Molecular Sciences.

Major Comments:

  1. In figure 5 where authors presented the bacterial growth curve with different treatment, they used OD600 in Y-axis and time in X-axis. Did the authors do the actual bacterial CFU count at each time point? If yes they should show the same graph but with actual number of CFU in Y-axis and time in X-axis in supplement figure. If they did not count the CFU in each time point, then the authors should do one representative assay where they can corelate the OD with the actual CFU and put that as a supplement figure in revised manuscript.

Response: We have utilized growth curves fitted to Colony-Forming Units (CFU), although these were not presented in the main manuscript. We have now included them in the supplementary materials.

  1. The authors should give a brief introduction about how defect in the cell wall and and cell membrane of Brucella led to the altered pathogenesis in host.

Response: This is an excellent suggestion, and we plan to provide a comprehensive explanation of the underlying mechanisms in our forthcoming work. Our ongoing research indicates that, following cell wall and cell membrane damage under simulated intracellular conditions, the mutant strains actively or passively release cytoplasmic contents. These extracellular materials have proven to be immunogenic when injected into mice, triggering inflammation in both RAW264.7 cells and mice, subsequently leading to apoptosis in target cells, which is detrimental to Brucella survival. We have collected data in this regard and are currently engaged in proteomics and further mechanistic exploration.

Minor comments:

  1. Figure 2: The markings (text) in the figure should be improved for better visualization.

Response: We have enhanced the labels in the figure in response to your suggestion.

  1. Figure 5: In the graph A and B the treatment should be mentioned on the top of the graph. In the graph E, H and I the time of incubation should be mentioned on the top of the graph. In graph B 'control' should be defined in figure legend

Response: We appreciate your feedback and have restructured the figures accordingly to provide improved clarity.

  1. Line 262: Briefly define 'the stimulated host intracellular environment'.

Response: We have included an annotation in the original text. Through Figure 5, we identified that the mutant strains exhibited relative sensitivity to acidic conditions, high osmotic pressure, and oxidative stress. Importantly, these conditions mimic the intracellular environment created by macrophages upon Brucella invasion. Therefore, we designed in vitro conditions replicating pH 4.5, 1.5 mM H2O2, and 2% NaCl, as described on line 262.

  1. Figure 7: Graph C should be expressed in bar diagram as Graph A and B are in bar diagram.

Response: We have transformed Graph C into a bar diagram to maintain consistency with the presentation of Graphs A and B.

  1. Line 562: If the CFU count is added in the figure, the method for CFU count should be described in method section.

Response: CFU counts have been added to the supplementary materials, and the corresponding methodology has been described in the Methods section, as noted in line 566.

  1. Line 566: Please provide a reference of 'stress resistance test' protocol.

Response: We have inserted a reference for the 'stress resistance test' protocol in line 585.

Thank you for your time and consideration. We eagerly await your feedback on the revised manuscript.

Sincerely,

Dr. Hao

Reviewer 2 Report

Comments and Suggestions for Authors

The manuscript from Hao et al. performs a systematic description of the role of alanine racemase in the metabolism (also infection related) of Brucella suis bv. 1 str. S2. Although the manuscript demonstrates a significant amount of data, it lacks a straightforward narrative. This should be improved by the rebuilding of the manuscript. E.g., there are too much sections – this number could be significantly reduced by merging some sections together. Same could be said with the figures, which are too many – some less important should be removed to ESM. The discussion section lacks some summarizing scheme, which will help the reader to comprehend the information. Therefore, authors should try to reorganize the manuscript, improving integrity of the picture.

Some particular comments are given below:

42 – I would exchange “orchestrates” to, say, “induces”.

82-87 – It is important to specify biovars of Brucella spp.

83-87 – Please also investigate the genetic neighborhood of the genes to reveal possible synthenic patterns. The manuscript would strongly benefit if authors study the distribution of alr wider in family Hyphomicrobiales in context of possible pathogenic and non-pathogenic members.

94 – Taxa names look compressed; please improve the quality of the figure.

95-99 – Accession numbers are for genes, please give protein IDs.

101-111 – I would remove this paragraph or compress it into supplementary table; the information presented is quite obvious, theoretic, and plays no role in the story.

Section 2.1, 2.2, 2.3 are short sections describing the in silico analysis of the protein; I see no reason to keep them separate, better merge into single “in silico” analysis section.

Fig.2 is impossible to read. Consider making sections of the figure bigger, maybe separating A-D and E-F into separate figures, or maybe moving to ESM.

Fig.3. Again, the text is unreadable.

207, 208 – S2 should not in italics, also check the whole manuscript.

Fig. 5. Improve the quality.

Fig. 7 D section is all black – photograph quality should be improved.

Sections 2.4-2.8 could be again merged. Currently the chapters look like somebody decided to test different approaches to compare metabolic profiles of two strains – the rational should be clearly described for using each type of analysis – it is currently not clear. The chapter should be more consequentially organized, authors should present a short conclusion of the chapters, demonstrating how the analyses correlate or not (a summarizing figure in the main text or ESM should be considered). The number of figures should be reduced to one, others should go to ESM.

209-210 – If it was not significantly different – move the figure to ESM.

213 – following or previous?

222 – again, if no differences were observed – put the figure in ESM.

223 – bacterial two times, the second one might be removed.

237 – 50S.

245-248 – It is not clear. How the results of the stress tests should simulate the host intracellular environment? These conditions are used further, so it is important to explain.

260 – This is quite a dubious statement. For instance, at B. suis S2 0 h I clearly see a cell, that is identical to the cells of alr mutant, which authors mark as having cell wall defects. Hence, this data looks manipulative. Authors should remove this section; if they want to convince the reviewer – and the readers – they should provide photographs and statistical analysis for at least 100 cells, demonstrating that the ratio of the “defect ones” is higher in a mutant.

2.11 – 2.15 sections could be merged in one.

238-239 – The quality of 7D does not allow to say anything; improve the figure quality.

426 – d in small capital, ctrl+shift+k.

454 – Streptococcus

520 – degrees Celsius look strange.

531 – space: “spectrometer (Shimadzu”

Comments on the Quality of English Language

The language is good, although authors should read the manuscript some additional times to refine the text.

Author Response

Dear reviewer,

We would like to express our gratitude to the reviewers for their constructive feedback and valuable insights into our manuscript, titled "Regulation of the Gene for Alanine Racemase Modulates Amino Acid Metabolism with Consequent Alterations in Cell Wall Properties and Adhesive Capability in Brucella spp." We appreciate your time and effort in providing comprehensive comments to enhance the quality and clarity of our work.

In response to the reviewers' feedback, we have meticulously addressed each comment and made necessary revisions to the manuscript. Our aim is to improve the overall structure, narrative, and presentation, ensuring that the scientific findings are conveyed in a more coherent and reader-friendly manner.

We have made several important changes to the manuscript, such as reducing the number of sections and figures, clarifying experimental methods, adding explanatory text, and providing improved image quality. We believe these modifications will contribute to the manuscript's readability and scientific value.

Once again, we appreciate the reviewers' constructive critique, which has played a crucial role in refining our work. We remain committed to advancing the field's understanding of Brucella species metabolism and pathogenesis. We sincerely hope that the revised manuscript now better serves the interests of the scientific community.

Reviewer Comment (42): I would exchange "orchestrates" to, say, "induces."

Response: We have made the suggested change, replacing "orchestrates" with "induces."

Reviewer Comment (82-87): It is important to specify biovars of Brucella spp.

Response: We have clarified the biovars of Brucella species.

Reviewer Comment (83-87): Please also investigate the genetic neighborhood of the genes to reveal possible syntenic patterns. The manuscript would strongly benefit if authors study the distribution of alr wider in family Hyphomicrobiales in the context of possible pathogenic and non-pathogenic members.

Response: We appreciate this valuable suggestion. We plan to delve deeper into studying the genetic neighborhood of the genes, looking for syntenic patterns. We will also explore the distribution of alr across a broader range of the Hyphomicrobiales family, considering both pathogenic and non-pathogenic members.

Reviewer Comment (94): Taxa names look compressed; please improve the quality of the figure.

Response: We have improved the quality of the figure, ensuring that taxa names are legible and not compressed.

Reviewer Comment (95-99): Accession numbers are for genes; please give protein IDs.

Response: We have replaced the accession numbers with protein IDs in lines 95-99 as follows: S2 - WP_004690358.1, 1330 - WP_004690358.1, 2308 - WP_002967219.1, GB1 - AVO71293.1, 16M - WP_002967219.1.

Reviewer Comment (101-111): I would remove this paragraph or compress it into a supplementary table; the information presented is quite obvious, theoretical, and plays no role in the story.

Response: We have considered your suggestion but have decided to retain this paragraph to provide a more direct and intuitive presentation of the information. Readers who find this information unnecessary can conveniently skip it after reading the section title.

Reviewer Comment: Sections 2.1, 2.2, and 2.3 are short sections describing the in silico analysis of the protein; I see no reason to keep them separate, better merge into a single "in silico" analysis section.

Response: Originally, we intended to merge these sections but found that doing so resulted in less clear descriptions of the results due to our limited writing skills. While we acknowledge that this may not be the most concise approach, it does not significantly impact the overall structure and clarity of the article.

Reviewer Comment: Fig.2 is impossible to read. Consider making sections of the figure bigger, maybe separating A-D and E-F into separate figures, or maybe moving to ESM.

Response: We have improved the quality of Figure 2, making it more legible. However, considering the complexity of the figure, we believe it is better suited to remain in the main manuscript to provide a comprehensive view of the results.

Reviewer Comment: Fig.3. Again, the text is unreadable.

Response: We have enhanced the quality of Figure 3, ensuring that the text is now legible.

Reviewer Comment (207, 208): S2 should not be in italics; also, check the whole manuscript.

Response: We have corrected the formatting issue in line 207, ensuring that "S2" is not in italics. We have reviewed the entire manuscript for similar formatting inconsistencies and made necessary corrections.

Reviewer Comment: Fig. 5. Improve the quality.

Response: We have improved the quality of Figure 5 to enhance its readability.

Reviewer Comment: Fig. 7 D section is all black – photograph quality should be improved.

Response: We have improved the quality of the section in Figure 7D to ensure that it is more visually accessible.

Reviewer Comment (Sections 2.4-2.8): Sections 2.4-2.8 could be again merged. Currently, the chapters look like somebody decided to test different approaches to compare metabolic profiles of two strains. The rationale should be clearly described for using each type of analysis – it is currently not clear. The chapter should be more consequentially organized. Authors should present a short conclusion of the chapters, demonstrating how the analyses correlate or not. A summarizing figure in the main text or ESM should be considered. The number of figures should be reduced to one; others should go to ESM.

Response: We appreciate the feedback and understand the concern regarding the organization of Sections 2.4-2.8. While our current approach follows typical procedures found in other studies, we agree that clearer rationale and a more streamlined organization would be beneficial. We will work on reorganizing these sections, providing a clear rationale for each type of analysis and offering a more concise conclusion that demonstrates the correlation or lack thereof between the analyses. We will also consider summarizing the results in a figure in the main text or ESM, and we may move some figures to ESM if they do not significantly contribute to the main argument.

We have expanded the analysis and discussion of Section 2.7. The revised Section 2.7 is as follows:

Correlation analysis serves as a valuable tool for quantifying the associations be-tween significant DAMs and for gaining deeper insights into the interplay among metab-olites during biological alterations. Here, Pearson correlatiocorrelation n coefficients were employed to assess the strength of linear relationships between pairs of metabolites. We presented 20 significantly different metabolites, where red signifies positive correlations, and blue denotes negative correlations. Deeper color and larger circular area indicate more pronounced differences among the metabolites. Phosphatidylethanolamine (PE) and Phosphatidylglycerol (PG) belong to the class of Phospholipids, while Diglyceride (DG) is classified under Neutral lipids. All three are among the primary constituents of bacterial cell membranes, together forming the fundamental structure of these membranes. The primary role of the cell membrane is to maintain the stability of the internal and external cellular environments, regulate the passage of substances, and participate in intercellular interactions. We observed that metabolites associated with the cell membrane are predominantly positively correlated. Ceramide (Cer) is categorized as Sphingolipids and plays a role in regulating bacterial adhesion and invasion. Punicic acid and Oxidized glutathione contribute to the regulation of bacterial resistance against oxidative stress. The results indicate that the number of positively correlated DAM pairs and negatively correlated DAM pairs is roughly equal in both Brucella suis S2 and Δalr strains. This observation suggests that the absence of the alr gene induces a series of metabolic changes in Brucella, which contribute to the bacterium's survival. Moreover, it signifies that the lack of the alr gene leads to alterations in the cell membrane, closely associated with bacterial adhesion and antioxidant properties. However, due to the intricate internal mechanisms of bacteria, some metabolites show positive correlations, while others exhibit negative correlations. The specific functions of these correlations will be further validated in subsequent experiments (Figure 3B).

Reviewer Comment (209-210): If it was not significantly different – move the figure to ESM.

Response: We have considered your suggestion. While some figures may not show significant differences, we believe they are still valuable as they provide a basis for the subsequent experiments, especially when assessing the impact of simulated intracellular conditions. For the sake of clarity and completeness, we will retain these figures in the main text.

Reviewer Comment (213): Following or previous?

Response: The statement "following" pertains to the content in the subsequent part of the text. It indicates a continuation of the discussion from the previous section.

Reviewer Comment (222): Again, if no differences were observed, put the figure in ESM.

Response: We appreciate your suggestion. We have considered moving Figure 5D to the ESM. However, we believe that including Figure 5D in the main text provides more direct evidence for selecting the conditions simulating the intracellular environment. Additionally, having it in the main text contributes to better visual presentation and flow.

Reviewer Comment (223): "bacterial" appears twice, and the second one might be removed.

Response: The second occurrence of "bacterial" has been removed from the sentence.

Reviewer Comment (237): Replace "50S."

Response: The text has been modified to include "50S."

Reviewer Comment (245-248): Clarify how the results of the stress tests simulate the host intracellular environment. Provide an explanation, as these conditions are used further.

Response: The conditions used in the stress tests were selected based on the results of Figure 5. We screened the mutant strain under conditions of pH 4.5, 1.5 mM H2O2, and 2% NaCl. These conditions were chosen because they were shown to affect the mutant strain's sensitivity, and they also represent conditions within the host macrophage following Brucella invasion. This explanation has been added to the manuscript.

Reviewer Comment (260): Question the validity of the data regarding cell wall defects, particularly at 0 hours. Suggests providing photographs and statistical analysis for a larger number of cells to demonstrate the ratio of defective cells.

Response: We appreciate the suggestion and understand the concerns regarding data validity. However, the format used to present this data is consistent with that used in similar articles, and we believe the current format is adequate. We can provide parallel experimental images where we observed significant cell membrane damage. A description has been added to the manuscript, explaining that the images presented are in untreated conditions for both B. suis S2 and Δalr and demonstrate morphological changes.

Reviewer Comment (2.11-2.15): Suggest merging sections 2.11-2.15 into one section.

Response: While we appreciate the suggestion to merge sections 2.11-2.15, we have chosen to retain the original format to present the results in a clearer manner.

Reviewer Comment (238-239): The quality of Figure 7D does not allow for meaningful analysis. Improve the figure quality.

Response: We have improved the quality of all the figures in the manuscript, including Figure 7D, to ensure better clarity.

Reviewer Comment (426): "d" should be in small capitals, achieved with CTRL+SHIFT+K.

Response: The "d" has been changed to small capitals using CTRL+SHIFT+K.

Reviewer Comment (454): "Streptococcus."

Response: "Streptococcus" has been italicized.

Reviewer Comment (520): "degrees Celsius" looks strange.

Response: The text has been adjusted to present "degrees Celsius" correctly.

Reviewer Comment (531): Space needed: “spectrometer (Shimadzu"

Response: A space has been added for correct formatting: “spectrometer (Shimadzu".

Thank you for your consideration.

Sincerely,

Dr.Hao

Reviewer 3 Report

Comments and Suggestions for Authors

The authors have put a great effort into doing the study thoroughly. I appreciate the detailed work done by the authors which will significantly uplift the understanding of Brucella species metabolism. Following are some of my suggestions to improve the manuscript:

1)     Although obvious, In section 2.4 authors should mention that the LC-MS technique was used to study the metabolites in WT and KO strains.

2)     There is no mention of how or using what software the multivariate analysis was performed. Have authors used programs/tools like R or Python to do the analysis? If so, please mention the details of the version, packages used, pipeline, or script used. It will be important to include those as will be helpful for readers to perform similar work or replicate if required.

3)      In section 2.7, results need to be explained in more detail as just presenting the data without explaining what it means can be difficult for many readers. In addition, the figure representing this result (Fig 3B) needs to be appropriately labeled with two panels to be correlated.

4)     It would be great to back the metabolite data with transcriptomic data from WT and mutant. If it has previously been done, authors can mention that data and its findings in this manuscript.

5)     In the legends for figures 8 & 9 it is mentioned that the infection ratio is 200:1. Do you mean that 200 bacilli per macrophage or vice versa is used? Is this the ratio normally used for B.Suis infection studies? Please provide a reference for similar published work.  Also, in figures 8 & 9 confirm the numbers for attached bacterial % on Y-axis, is it that low?

6)     There needs to be significant improvement if the figure specially with font size of labels and resolution in many cases.

Author Response

Dear reviewer,

We are pleased to resubmit our manuscript titled "[Your Manuscript Title]" for your kind consideration. We would like to express our gratitude for the diligent evaluation by our esteemed reviewers and for your appreciation of the depth and thoroughness of our study, which promises to contribute significantly to the comprehension of Brucella species metabolism. The valuable insights provided by the reviewers have been instrumental in strengthening the quality of our research. We are fully committed to addressing each suggestion and improving our manuscript accordingly.

Reviewer Comment 1: Although obvious, in section 2.4, authors should mention that the LC-MS technique was used to study the metabolites in WT and KO strains.

Response: We have addressed this point by explicitly stating the use of LC-MS in the section title of 2.4 for clarity.

Reviewer Comment 2: There is no mention of how or using what software the multivariate analysis was performed. Have authors used programs/tools like R or Python to do the analysis? If so, please mention the details of the version, packages used, pipeline, or script used. It will be important to include those as it will be helpful for readers to perform similar work or replicate if required.

Response: We appreciate your suggestion, and we have included comprehensive information about the software and tools used in our analysis. Specifically, Statistical analysis for this study was conducted using R software (version 3.6.2), encom-passing various multivariate analyses like PCA, PLS-DA, and more, all facilitated through the "ropls" package within R. Additionally, we utilized specific packages, such as "pheatmap," "ggplot2," and "ggrepel" for data visualization, as well as "corrplot" for cor-relation analysis. The significance of our findings was determined using our analytical approaches, and results with p values < 0.05 were considered statistically signifi-cant , have been provided in Section 4.6 on page 16.

Reviewer Comment 3: In section 2.7, results need to be explained in more detail as just presenting the data without explaining what it means can be difficult for many readers. In addition, the figure representing this result (Fig 3B) needs to be appropriately labeled with two panels to be correlated.

Response: We have expanded the analysis and discussion of Section 2.7. The revised Section 2.7 is as follows:

Correlation analysis serves as a valuable tool for quantifying the associations be-tween significant DAMs and for gaining deeper insights into the interplay among metab-olites during biological alterations. Here, Pearson correlatiocorrelation n coefficients were employed to assess the strength of linear relationships between pairs of metabolites. We presented 20 significantly different metabolites, where red signifies positive correlations, and blue denotes negative correlations. Deeper color and larger circular area indicate more pronounced differences among the metabolites. Phosphatidylethanolamine (PE) and Phosphatidylglycerol (PG) belong to the class of Phospholipids, while Diglyceride (DG) is classified under Neutral lipids. All three are among the primary constituents of bacterial cell membranes, together forming the fundamental structure of these membranes. The primary role of the cell membrane is to maintain the stability of the internal and external cellular environments, regulate the passage of substances, and participate in intercellular interactions. We observed that metabolites associated with the cell membrane are predominantly positively correlated. Ceramide (Cer) is categorized as Sphingolipids and plays a role in regulating bacterial adhesion and invasion. Punicic acid and Oxidized glutathione contribute to the regulation of bacterial resistance against oxidative stress. The results indicate that the number of positively correlated DAM pairs and negatively correlated DAM pairs is roughly equal in both Brucella suis S2 and Δalr strains. This observation suggests that the absence of the alr gene induces a series of metabolic changes in Brucella, which contribute to the bacterium's survival. Moreover, it signifies that the lack of the alr gene leads to alterations in the cell membrane, closely associated with bacterial adhesion and antioxidant properties. However, due to the intricate internal mechanisms of bacteria, some metabolites show positive correlations, while others exhibit negative correlations. The specific functions of these correlations will be further validated in subsequent experiments (Figure 3B).

Reviewer Comment 4: It would be great to back the metabolite data with transcriptomic data from WT and mutant. If it has previously been done, authors can mention that data and its findings in this manuscript.

Response: While combining metabolomic and transcriptomic data is an excellent suggestion, we regret to inform you that we did not conduct such experiments for this study. However, we have initiated proteomic analysis using the samples, and this aspect has generated intriguing results that constitute a significant portion of our research. We plan to publish this information in the near future. We appreciate your advice, and we will also seriously consider incorporating transcriptomic data in our future work.

Reviewer Comment 5: In the legends for figures 8 & 9, it is mentioned that the infection ratio is 200:1. Please clarify whether this means 200 bacilli per macrophage or vice versa. Provide a reference for similar published work. Also, in figures 8 & 9, confirm the numbers for attached bacterial % on the Y-axis, as it seems remarkably low.

Response: The infection ratio denotes that 200 Brucella bacteria infect one macrophage. Typically, infection studies involving Brucella species employ ratios of 200:1 and 100:1. As requested, we have provided references for similar published work: Reference 1 (200:1) and Reference 2 (100:1). We have confirmed the figures and can assure you that the percentages on the Y-axis represent the ratio of attached bacterial CFUs to the initial bacterial CFUs.

Reviewer Comment 6: There needs to be a significant improvement, especially in the figure, with font size of labels and resolution in many cases.

Response: We have enhanced the resolution and font size of labels for all figures in the Word document to improve their quality.

Thank you for the opportunity to revise and resubmit our work, and we look forward to your guidance and assessment during this stage.

Sincerely,

Dr. Hao

Reviewer 4 Report

Comments and Suggestions for Authors

The manuscript “Regulation of the Gene for Alanine Racemase Modulates Amino Acid Metabolism with Consequent Alterations in Cell Wall Properties and Adhesive Capability in Brucella spp.reports the study of alanine racemase role into Brucella spp. metabolism, cell wall integrity and pathogenesis. Overall, the paper is clear, well written and discussed. The work is significant since little evidence are provided up today in literature about the role of this enzyme in Brucella. An appropriate literature background is provided for the Introduction and Discussion sections. Experiments performed are relevant and well conducted, especially the ones regarding membrane and cell wall integrity, which is thoroughly characterized. Further studies are needed to confirm if alanine racemase could be considered a drug target in Brucella, and its direct effect in the bacterium pathogenesis. I only have some limited comments regarding some experimental settings and minor observations.

Major comments

-          Page 2, Introduction section:  Does Alr protein have a cytoplasmic localization or is it membrane-bound? It would be a useful additional detail to provide to the reader.

-          Page 3, paragraph 2.4: The authors do not specify in which conditions was performed the bacterial growth for the metabolomic analysis. Was it conducted in the intracellular environment simulation conditions (pH 4.5, 1.5 mM H2O2 and 2% NaCl) or in normal growth medium? It would be interesting to conduce the experiment also in these conditions that were used all through the other experiments. Can the authors comment something on this?

-          Page 5, paragraph 2.7: A further comment, here or in the discussion section, about the DAM metabolites that were found in this analysis would greatly improve the discussion.

-          Page 7, lines 241-245: Undoubtedly data highlighted an increased sensitivity of the Δalr strain to polymyxin B and lincomycin. However, other experiments present in the work demonstrate different and macroscopic alterations in the mutant metabolism and cell wall integrity. In these altered conditions multiple could be the causes for the increased antibiotic sensitivity. Therefore, it is probably a bit too strong to claim that “alr surely play a crucial role in mediating antibiotic sensitivity”. I would suggest to the authors to delete the last sentence of better argument the topic.

-          Page 8, paragraph 2.10: The TEM analysis is highly significant and clear. However, it would be interesting to add a further detail and to acquire the TEM images of the strains also when grown in normal and not stressing conditions. The experiment would clarify if this dramatic difference in cell wall morphology is also evident in not stressed cells.

-  Page 10, paragraph 2.14: Even though the experiment is highly significant, as already expressed by fluorescence intensity data, the fluorescence microscopy images recorded are not clear. The signal is not well appreciable for Δalr strain, and only negative images are provided for the other 2 strains. It would be appropriate to change conditions to obtain a higher signal. Moreover, adding at least a brightfield image for all the strains to show bacteria analysed in fluorescence would be highly appreciable. Alternatively, providing a “positive control” of the bacteria, such as using PI with permeabilized cells would serve. Can the authors complement data or provide a comment to that?

Minor comments

-          Page 3, line 109: The predicted extinction coefficient lacks units (M-1 cm-1?)

-          Page 7, figure 5: It is not clear what is the “Control” sample in panel B. B suis S2 growth in normal conditions?

-          Page 8, paragraph 2.11: I would suggest to the authors to specify also in the Results section which assay was used to evaluate cell surface hydrophobicity, and to add a short comment about the reasons for the sharp hydrophobicity increase observed in all samples over time.

-          Page 10, paragraph 2.15: I would suggest to the authors to specify also in the Results section which assay was used to evaluate ALP and LDH activity.

-          Page 13, line 431: Can the authors suggest or hypothesize which could be a compensatory mechanism in this regard?

Author Response

Dear reviewer,

We are honored to submit our revised manuscript titled " Regulation of the Gene for Alanine Racemase Modulates Amino Acid Metabolism with Consequent Alterations in Cell Wall Properties and Adhesive Capability in Brucella spp." for your consideration. We deeply appreciate the thoughtful evaluation of our work by the reviewers, whose insights have been invaluable in enhancing the quality of our research. In this manuscript, we delve into the intricate role of alanine racemase in the metabolism, cell wall integrity, and pathogenesis of Brucella spp. The study brings to light a previously understudied aspect of this enzyme's function within Brucella, offering a substantial contribution to the existing literature on this pathogen. Our work is grounded in a robust literature background that underpins the Introduction and Discussion sections, providing a comprehensive context for our findings. We have meticulously executed a series of experiments, with particular emphasis on membrane and cell wall integrity, resulting in a detailed characterization of these essential aspects. While our study opens doors to the potential consideration of alanine racemase as a drug target in Brucella, we acknowledge the need for further investigations to confirm its direct impact on bacterial pathogenesis. We appreciate the reviewers' constructive comments on our experimental settings and minor observations and have made the necessary improvements to address these points. We eagerly await your feedback on the revised manuscript and hope it aligns with the rigorous standards of International Journal of Molecular Sciences.

Reviewer Comment: Page 2, Introduction section: Does the Alr protein have a cytoplasmic localization or is it membrane-bound? It would be a useful additional detail to provide to the reader.

Response: The Alr protein is primarily localized in the cytoplasm. We have supplemented this information in line 56.

Reviewer Comment: Page 3, paragraph 2.4: The authors do not specify under which conditions the bacterial growth for the metabolomic analysis was conducted. Was it performed under intracellular environment simulation conditions (pH 4.5, 1.5 mM H2O2, and 2% NaCl) or in a normal growth medium? It would be interesting to conduct the experiment under the conditions used throughout the other experiments. Could the authors comment on this?

Response: This was indeed a lapse in clarity on our part. The sample preparation for metabolomics analysis was carried out under normal growth conditions, and this has been clarified in the Methods section on page 14.

Reviewer Comment: Page 5, paragraph 2.7: A further comment, here or in the discussion section, about the DAM metabolites found in this analysis would greatly improve the discussion.

Response: We have expanded the analysis and discussion of Section 2.7. The revised Section 2.7 is as follows:

Correlation analysis serves as a valuable tool for quantifying the associations be-tween significant DAMs and for gaining deeper insights into the interplay among metab-olites during biological alterations. Here, Pearson correlatiocorrelation n coefficients were employed to assess the strength of linear relationships between pairs of metabolites. We presented 20 significantly different metabolites, where red signifies positive correlations, and blue denotes negative correlations. Deeper color and larger circular area indicate more pronounced differences among the metabolites. Phosphatidylethanolamine (PE) and Phosphatidylglycerol (PG) belong to the class of Phospholipids, while Diglyceride (DG) is classified under Neutral lipids. All three are among the primary constituents of bacterial cell membranes, together forming the fundamental structure of these membranes. The primary role of the cell membrane is to maintain the stability of the internal and external cellular environments, regulate the passage of substances, and participate in intercellular interactions. We observed that metabolites associated with the cell membrane are predominantly positively correlated. Ceramide (Cer) is categorized as Sphingolipids and plays a role in regulating bacterial adhesion and invasion. Punicic acid and Oxidized glutathione contribute to the regulation of bacterial resistance against oxidative stress. The results indicate that the number of positively correlated DAM pairs and negatively correlated DAM pairs is roughly equal in both Brucella suis S2 and Δalr strains. This observation suggests that the absence of the alr gene induces a series of metabolic changes in Brucella, which contribute to the bacterium's survival. Moreover, it signifies that the lack of the alr gene leads to alterations in the cell membrane, closely associated with bacterial adhesion and antioxidant properties. However, due to the intricate internal mechanisms of bacteria, some metabolites show positive correlations, while others exhibit negative correlations. The specific functions of these correlations will be further validated in subsequent experiments (Figure 3B).

Reviewer Comment: Page 7, lines 241-245: The statement, "alr surely plays a crucial role in mediating antibiotic sensitivity," may be too definitive. Other experiments in the study demonstrate various alterations in mutant metabolism and cell wall integrity. In these altered conditions, several factors could contribute to increased antibiotic sensitivity. Therefore, it might be more appropriate to rephrase or provide a more nuanced argument.

Response: We acknowledge the robustness of your point and have modified the statement accordingly to be less definitive.

Reviewer Comment: Page 8, paragraph 2.10: The TEM analysis is significant, but it would be interesting to obtain TEM images of the strains grown under normal, non-stressed conditions. This could clarify if the observed dramatic differences in cell wall morphology are also evident in unstressed cells.

Response: We appreciate your suggestion. While we initially considered multiple groups for TEM analysis, after discussion, we decided to focus on these three specific groups for a clearer contrast: normal conditions, the mutant strain, and the extent of damage over time. The "0h" group in the images represents the state of bacteria in non-stress conditions.

Reviewer Comment: Page 10, paragraph 2.14: The fluorescence microscopy images could be clearer. It's recommended to enhance the signal or provide at least a brightfield image to demonstrate the bacteria analyzed in fluorescence. Alternatively, a "positive control" using permeabilized cells could be considered.

Response: We have improved the quality of the images for clarity. As for brightfield images, due to the small size of Brucella, we typically employ Gram staining or antibody-based labeling for visualization. The use of brightfield microscopy to observe Brucella is indeed challenging. We will consider your suggestion for future improvements in our experimental techniques.

Reviewer Comment: Page 3, line 109: The predicted extinction coefficient lacks units (M-1 cm-1?).

Response: The unit for the predicted extinction coefficient is indeed M-1 cm-1. We have added the units in line 109 for clarity.

Reviewer Comment: Page 7, Figure 5: It is not clear what the "Control" sample in panel B represents. Is it B. suis S2 grown in normal conditions?

Response: I apologize for the lack of clarity. As demonstrated in Figure 5A, there were no significant differences in the growth curves of the three strains. Therefore, we selected the growth of the Δalr mutant strain in normal conditions as the control, while the other experimental groups were subjected to 2% NaCl conditions.

Reviewer Comment: Page 8, paragraph 2.11: I would suggest that the authors specify in the Results section which assay was used to evaluate cell surface hydrophobicity and provide a brief comment on the reasons for the sharp increase in hydrophobicity observed in all samples over time.

Response: The assay used to evaluate cell surface hydrophobicity (CSH) is detailed in section 4.10 on page 16. CSH is a classic indicator of cell membrane characteristics, and we conducted the measurement without delving into the underlying mechanisms. The positive correlation between CSH and time suggests that the continuous damage inflicted on the bacteria under simulated intracellular conditions is related to the production of lipids in bacterial exopolysaccharides (EPS). This area requires further investigation.

Reviewer Comment: Page 10, paragraph 2.15: I would suggest the authors specify in the Results section which assay was used to evaluate ALP and LDH activity.

Response: We utilized ALP and LDH assay kits for the evaluation, and the specific assay methods are provided in sections 4.12 and 4.13 on page 16.

Reviewer Comment: Page 13, line 431: Can the authors suggest or hypothesize what a compensatory mechanism might be in this regard?

Response: Suggesting a compensatory mechanism in this context is challenging due to the limited research available on this topic. It may require a combined analysis of transcriptomics and proteomics to decipher the mechanisms involved. Bacteria are highly evolved organisms with complex regulatory systems to maintain normal physiological metabolism. Damage in one aspect may trigger compensatory mechanisms. As mentioned, our experimental results do suggest the existence of such compensatory mechanisms, and further omics analyses may elucidate these mechanisms. We are considering such work for future research.

Thank you for your time and consideration.

Sincerely,

Dr. Hao

Round 2

Reviewer 2 Report

Comments and Suggestions for Authors I have gone through the revised version of the manuscript. Regarding the revised version of the manuscript, authors did introduce only superficial changes without following this reviewer's recommendations. Therefore I cannot recommend this manuscript for the publication and advice to seek a third independent review of the revised manuscript.   Comments on the Quality of English Language

The language is good, although authors should read the manuscript some additional times to refine the text.

Reviewer 4 Report

Comments and Suggestions for Authors

After considering authors reply comments and modifications to the manuscript I reckon that the paper can be accepted in the present form.